# Translational Frameshifting in the *chlD* Gene Gives a Clue to the Coevolution of the Chlorophyll and Cobalamin Biosyntheses

**DOI:** 10.3390/microorganisms10061200

**Published:** 2022-06-11

**Authors:** Stepan Kuznetsov, Alexander Milenkin, Ivan Antonov

**Affiliations:** 1Moscow Institute of Physics and Technology, 141701 Dolgoprudny, Russia; stepan.v.kuznetsov@phystech.edu (S.K.); milenkin.aa@phystech.edu (A.M.); 2Institute of Bioengineering, Research Center of Biotechnology, Russian Academy of Science, 117312 Moscow, Russia; 3Laboratory of Bioinformatics, Faculty of Computer Science, National Research University Higher School of Economics, 101000 Moscow, Russia

**Keywords:** chlorophyll, cobalamin, vitamin B12, frameshifting, evolution, chelatase, chlIDH, cobNST

## Abstract

Today, hundreds of prokaryotic species are able to synthesize chlorophyll and cobalamin (vitamin B12). An important step in the biosynthesis of these coenzymes is the insertion of a metal ion into a porphyrin ring. Namely, Mg-chelatase ChlIDH and aerobic Co-chelatase CobNST are utilized in the chlorophyll and vitamin B12 pathways, respectively. The corresponding subunits of these enzymes have common evolutionary origin. Recently, we have identified a highly conserved frameshifting signal in the *chlD* gene. This unusual regulatory mechanism allowed production of both the small and the medium chelatase subunits from the same gene. Moreover, the *chlD* gene appeared early in the evolution and could be at the starting point in the development of the chlorophyll and B12 pathways. Here, we studied the possible coevolution of these two pathways through the analysis of the chelatase genes. To do that, we developed a specialized Web database with comprehensive information about more than 1200 prokaryotic genomes. Further analysis allowed us to split the coevolution of the chlorophyll and B12 pathway into eight distinct stages.

## 1. Introduction

The early evolution of photosythesis has been an intriguing but complex research subject for many years [1,2]. Clearly, the appearance of photosythesis required the ability of the ancient prokaryotes to synthesize chlorophyll (CHL). Chlorophyll is a relatively complex organic compound that includes Mg^2+^ ion in its tetrapyrrole ring [3]. Modern organisms utilize a number of different enzymes to produce chlorophyll from a metal-free tetrapyrrole intermediate Uroporphyrinogen III (Uro) [4]. On the other hand, other species utilize several alternative pathways that can turn Uro into cobalamin (vitamin B12), heme, or coenzyme F430 (Figure 1). These coenzymes contain different metal ions in their tetrapyrrole rings [5]. Namely, cobalamine (vitamin B12), haem, and coenzyme F430 include Co^2+^, Fe^2+^ and Ni^2+^, respectively. The enzymes that insert those ions into the corresponding organic rings are called chelatases.

The biosynthesis of Mg^2+^-containing chlorophylls and bacteriochlorophylls requires the magnesium chelatase that performs the insertion of the magnesium ion (Mg^2+^) into protoporphyrin IX [6]. This enzyme consists of small (I), medium (D), and large (H) subunits [7,8]. The corresponding genes are called *chlI*, *chlD*, and *chlH* in the genomes of chlorophyll-producing organisms and *bchI*, *bchD*, and *bchH* in bacteriochlorophyll producers. To reduce the number of different gene names in this work, we will use the *chlI*, *chlD*, and *chlH* terminology for both groups of genes.

Interestingly, the cobalamin biosynthesis pathway has two alternative routes—the anaerobic and aerobic ones (Figure 1) [9]. Strict anaerobes only posses the anaerobic B12 pathway. On the other hand, cobalamin-producing facultative anaerobes or aerobic organisms usually possess both routes and switch between them depending on the current living conditions [10]. From the evolutionary point of view, it is logical to assume that the aerobic route could only evolve when free oxygen was available in the Earth’s atmosphere, e.g., only after the CHL pathway and oxygenic photosynthesis have already existed. Moreover, the interplay between the CHL pathway and the aerobic B12 route is not limited to the oxygen itself. It has been well established that the cobalt chelatase CobNST from the aerobic route resembles the magnesium chelatase ChlIDH as it also consists of the small, medium and large subunits encoded by *cobS*, *cobT*, and *cobN* genes, respectively [11]. Additionally, the *chlI* and *chlD* from the chlIDH chelatase are likely to be homologous to the *cobS* and *cobT* genes of the cobNST chelatase [12]. The similarity between the large subunit genes (i.e., *chlH* and *cobN*) has also been reported [13].

Later studies have revealed additional existing strategies to encode the cobNST enzyme. First, Rodionov et al. [14] have suggested that in some species, the products of the *chlI* and *chlD* can replace the *cobS* and *cobT* genes and function as the small and medium subunits of the aerobic Co-chelatase. Moreover, a functional programmed ribosomal frameshifting (PRF) found in some *chlD* genes may allow the production of both the small and the medium subunits of the cobNST chelatase from the same gene [15,16]—see Appendix A. Altogether, this indicates a close connection between the CHL pathway and the aerobic B12 route suggesting their possible coevolution [17].

Here, we attempted to organize all the different strategies to encode cobNST and chlIDH enzymes and suggest a logical model of the CHL and aerobic B12 pathways coevolution. Our strategy was to pinpoint possible prokaryotic genomes that may still be using the ancient strategies to encode Mg- and/or Co-chelatases. Such “living fossil” species can be used to reconstruct the early stages of the CHL and B12 pathway evolution.

## 2. Materials and Methods

The Chelatase DB was developed using the Django framework implemented in Python. To populate the database the tBLASTn [18] tool was used to identify the genomes containing *chlD* gene(s). The translations of the three *chlD* genes with the validated frameshifting signals were used as queries [16]. To allow the identification of the other frameshifted *chlD* genes, two separate tBASTn searches (E-value threshold $=10^{-6}$) were performed using the N-terminal (i.e., before the frameshift) or the C-terminal (after the frameshift) parts of each query protein. The genomic regions with adjacent hits (in the correct order) with up to one frameshift were classified as *chlD* genes.

In order to identify the small and the large chelatase subunit genes as well as the other genes from the chlorophyll and cobalamin biosynthesis pathways, tBLASTn search was performed for a set of reference query proteins. Given the similarities between different chelatase subunits, we imposed additional constraints on their lengths. Namely, we required that the small chelatase subunits were between 250 aa and 500 aa, medium were between 500 aa and 800 aa, and large were between 1000 aa and 2000 aa (these values corresponded to the lengths of the annotated chelatase subunits). The identified genes were automatically annotated (i.e., assigned a gene name such as *chlD*, *bchD*, or *cobT*) by the best reciprocal hit approach.

We used the identified representative sets of the genes from the CHL and B12 pathways to predict whether a given prokaryotic genome possessed the corresponding biochemical pathway. Additionally, the automatically assigned genotype of the species mentioned in the text was double-checked with the information from the KEGG database (the “Porphyrin metabolism” section) [19]. The phylogenetic tree was constructed using RAxML version 8.2.12 [20]. Subsequent phylogenetic analyses and visualizations were performed using the “ape” R package [21]. The RNA secondary structures were predicted using the RNAfold web server [22] available at http://rna.tbi.univie.ac.at/cgi-bin/RNAWebSuite/RNAfold.cgi (accessed on 1 April 2022).

All the data files and scripts used to generate figures and tables from the article are available at https://github.com/vanya-antonov/article-chelatase-db (accessed on 1 April 2022).

## 3. Results

Our recent analysis [16] as well as earlier studies [8,14,23] have suggested that there were several different strategies to encode aerobic cobalt chelatase cobNST. Interestingly, in a number of species, the products of the Mg-chelatase genes (*chlD* and/or *chlI*) functioned as corresponding subunits of the cobNST complex (Appendix A). To explore the diversity of the strategies utilized by different prokaryotes to encode chelatases, we developed a specialized Web database called Chelatase DB (http://ivanya.com/chelatase/ (accessed on 1 April 2022)). For each prokaryotic genome, the Chelatase DB provided information about the number of the small, medium, and large subunit genes, the existence of the chlorophyll and/or B12 biosynthesis pathway and reference to the KEGG database for detailed analysis. Additionally, the database contained a collection of the frameshifting signals predicted in some *chlD* genes.

To identify the possible starting point in the evolution of the CHL pathway, the *chlD* gene encoding medium subunit of the Mg-chelatase was analyzed. We chose this gene because it was likely to appear early in the evolution, since it may have been present in the genome of the last universal common ancestor (LUCA) [24]. Thus, it was possible that the modern genes encoding medium chelatase subunits were descendants of the putative *chlD* gene from LUCA. Our previous analysis [16] of the frameshifted *chlD* (fs-*chlD*) genes has demonstrated that such genes were present in all major prokaryotic phyla (including archaea, proteobacteria, and actinobacteria)—Appendix A. Indeed, on the phylogenetic tree of 286 medium subunit genes all the fs-*chlD* genes were located near the root of the tree (Appendix A). Thus, the modern *chlD*, *bchD*, and *cobT* genes may have originated from the ancient fs-*chlD* gene.

Our next goal was to use the functionality of the Chelatase DB to reveal a possible evolutionary trajectory of the CHL and B12 pathways with respect to the corresponding chelatase genes. Namely, we inspected different chlIDH/cobNST genotypes and matched them with the corresponding predicted phenotypes (i.e., the ability to synthesize CHL and/or B12). This allowed us to define eight stages in the evolution of CHL and B12 pathways as well as *chlD* gene—see Table 1. Since fs-*chlD* gene had an ancient origin, it was logical to assume that it appeared in the evolution before the complete CHL and B12 pathways existed. Therefore, we search the Chelatase DB for the genomes with fs-*chlD* genes that did not have the genes from the CHL and B12 pathways. Indeed, we identified several such species including *Spirochaeta thermophyla* and *Brevefilum fermentans* (Table 1, stage 1). All of these bacteria that corresponded to the conditions of the prechlorophyll era were anaerobic [25]. Moreover, according to the global prokaryotic phylogentic tree, both Spirochaetes and Chloroflexi were among the most ancient prokaryotic phyla that are present today [26]. The presence of the frameshifting signals in the fs-*chlD* genes from *S. thermophyla* and *B. fermentans* suggested that they produced two proteins from the same mRNA (Appendix A). In the *chlD* genes from various taxa −1 frameshifts have been observed more frequently than +1 frameshifts [16]. The recoding signals in these genes usually consisted of AT-rich “slippery sites” together with a stimulatory RNA secondary structures located downstream. More info about the frameshifting mechanism can be found elsewhere [27]. Due to the stochastic nature of the programmed frameshifting, ribosomes reading the same mRNA can produce two different proteins that allows cotranslational assembly of protein complexes. Consequently, it was likely that the two proteins produced from fs-*chlD* mRNA interacted with each other—similarly to the CobS and CobT subunits [28]. However, the function of this protein complex was unclear since the genomes of the corresponding bacteria completely lacked the CHL and B12 pathways as well as the large chelatase subunit gene (either *chlH* or *cobN*). Together, these observations indicated that the *S. thermophyla* and *B. fermentans* could be living prokaryotic fossils and their genotypes can be considered as the starting point in the evolution of the CHL and B12 pathways.

It has previously been reported that the *chlI* gene encoding the small chelatase subunit was homologous to the N-terminal part of the *chlD* gene [29]. Therefore, the next putative evolutionary event was duplication of the N-terminal part of the fs-*chlD* gene and formation of a separate *chlI* gene. In this case, the frameshifting signal in the fs-*chlD* was likely to disappear since there was no need to produce both proteins from a single gene. Accordingly, we found another prokaryotic group that had *chlD* and *chlI* genes but still lacked the CHL and B12 pathways (Table 1, stage 2). Interestingly, the ability of the cotranslational interaction between the products of these proto-*chlI* and proto-*chlD* genes was possible since they were located next to each other on the genome and can be present in the same polycistronic mRNA. Although the function of the proteins produced from these two genes was still unclear, the corresponding genotypes can be considered as the next stage in the early evolution of CHL biosynthesis. Moreover, we identified several Chloroflexi species (from the Anaerolineaceae group) with a potential intermediate genotype between Stage 1 and 2. Namely, *Anaerolinea thermophila UNI-1*, *Ornatilinea apprima*, *Longilinea arvoryzae*, and *Levilinea saccharolytica* possessed the fs-chlD genes with characteristic frameshifting signals but also had a separate *chlI* gene. Interestingly, these species were also strict anaerobes and thermophiles suggesting their ancient origin [30]. Thus, there were a number of living prokaryotes that provided insights into the early evolution of the modern Mg-chelatase enzyme.

A good representative case of the stage 3 was photosynthetic proteobacteria *Polynucleobacter duraquae* that had the *chlH* gene encoding the large subunit of the Mg-chelatase (Table 1, stage 3). Similar to the genomes from stage 2, all three chelatase subunit genes were located at the same genomic locus, possibly allowing cotraslational assembly of the Mg-chelatase from a polycystronic mRNA. According to the KEGG database, *P. duraquae* possessed all the major genes required for CHL biosynthesis. It was likely that this bacteria was only capable of anoxygenic photosynthesis [31] supporting its relatively early position in our proposed evolutionary system.

At stage 4, we placed anoxygenic proteobacteria [32] that in addition to chlorophyll biosynthesis also had the B12 pathway (Table 1, stage 4). These bacteria possessed anaerobic B12 pathway as well as many genes from the aerobic route. However, it was unclear if they were able to synthesize cobalamin using oxygen since these species were anaerobic. Additionally, their genome did not encode the large subunit of the aerobic Co-chelatase (the *cobN* gene). Thus, if the B12 biosynthesis was possible through the aerobic route, then the Mg-chelatase from these species might be able to function as a Co-chelatase as well.

Modern Cyanobacteria (such as *Synechococcus elongatus*) were at the next stage in our model because they were able to perform oxygenic photosynthesis, encoded large subunits for both Mg- and Co-chelatases, and could produce cobalamin via anaerobic and aerobic routes (Table 1, stage 5). Interestingly, Cyanobacteria did not have the *cobS* and *cobT* genes that encode the small and the medium subunits of the cobNST chelatase. It has been suggested that the corresponding subunits from the Mg-chelatase (*chlI* and *chlD*) could substitute them [14].

A number of oxygenic phototrophic Proteobacteria had the most advanced genotype with respect to the Mg- and Co-chelatases (Table 1, stage 6). In comparison to Cyanobacteria, their genomes contained *cobS* and *cobT* genes that were likely duplicated from the *chlI* and *chlD* and subsequently specialized to function as subunits of the cobNST chelatase. Since many of these bacteria (e.g., *Rhodospirillum rubrum*) were able to live under both aerobic and aerobic conditions, their genomes encoded both routes of B12 biosynthesis [33].

Starting from stage 1 to stage 6, we observed gradual development of the gene sets encoding chlIDH and cobNST chelatases. On the other hand, a number of modern heterotrophic Actinobacteria (such as *Mycobacterium tuberculosis*) did not have the CHL pathway but encoded both pathways of B12 biosynthesis (Table 1, stage 7). However, they utilized the *chlI* and *chlD* genes, rather than *cobN* and *cobS*, to encode the small and the medium subunits of their aerobic Co-chelatase. This suggested that their chelatase genotypes originated from the Cyanobacteria where *chlI* and *chlD* genes performed similar function. Thus, we hypothesized that Actinobacteria from stage 7 adopted some genes from the Cyanobacterial CHL pathway (stage 5) and lost the rest of the CHL biosynthesis genes. Finally, stage 8 represented further simplification of the chelatase genotype (Table 1, stage 8) where a frameshifting signal in the fs-*chlD* gene allowed the production of both the small and the medium subunits of the Co-chelatase for the aerobic route of B12 biosythesis [16].

## 4. Discussion

In this paper, we presented a model for the possible coevolution of the CHL and B12 (aerobic route) pathways (Figure 2). To develop this model, we assumed that among the thousands of the sequenced prokaryotic genomes, there were examples of so-called “living fossils”, i.e., the direct descendants of the species from the early days of cellular life on Earth. Our analysis was based on the *chlD* gene due to its likely early appearance in the evolution [24]. We hypothesized that the frameshifted version of this gene was the starting point in the early evolution of CHL pathway. Namely, we identified this gene in several bacteria that could be considered as living fossils (stage 1). We also identified several other bacteria where a part of *chlD* gene was probably duplicated producing the proto-*chlI* gene (stage 2). Importantly, the stage 1 and 2 phlyla (Spirochaetes, Chloroflexi, Thermodesulfobacteria, and Nitrospirae) that appeared in the evolution before the Proteobacteria from the stage 3 and 4 [26] supporting our suggested evolutionary order. Interestingly, species from stage 1 and 2 did not have B12 or CHL pathways as well as either large subunit genes. On the other hand, the fs-*chlD*, *chlI* and *chlD* genes from these species can produce two proteins that may interact with each other and form a separate complex [28]. To our knowledge, the function of the *chlD* + *chlI* complex (without the a subunit protein) has not been studied and its function is unknown. Thus, further studies of these genomes may reveal the new function for the corresponding proteins.

The appearance of the CHL and the aerobic B12 pathways as well as oxygenic photosynthesis was observed in Cyanobacteria at stage 5 of our system. Clearly, the aerobic B12 pathway could only appear after the oxygenic photosynthesis; however, we were not able to find any oxygenic phototroph without the aerobic B12 pathway. Although we believe that such putative bacteria existed at some evolutionary point, they probably adopted the more efficient aerobic B12 pathway after its appearance and therefore could not be found today. Proteobacteria at stage 6 represented the most advanced chelatase genotypes where all the subunits of the chlIDH and cobNST enzymes were encoded by specialized genes. On the other hand, a possible reduction of the Cyanobacterial genomes could gave rise to the modern Actinobacteria that utilized *chlI* and *chlD* genes to encode small and medium cobNST subunits. Indeed, according to the global phylogenetic tree, Actinobacteria had a common ancestor with Cyanobacteria which appeared in the evolution first [26]. Finally, we observed that some prokaryotes utilized the ancient strategy to encode both the small and medium subunits of the aerobic Co-chelatase using frameshifting signal in the fs-*chlD* gene (stage 8). The *chlD* genes with frameshifting signals in these genomes likely originated via horizontal gene transfer rather than the signals evolving once again in the conventional *chlD*/*bchD* genes. This is supported by the fact the fs-*chlD* genes do not group with the other genes without frameshifts (Appendix A).

Another interesting question was related to the anaerobic cobalamin biosynthesis pathway—e.g., whether it appeared before or after the CHL biosynthesis pathway. Interestingly, the Chelatase DB included several archaeal species (such as *Methanobacterium paludis* and *Thermoplasma acidophilum*) that contained the *chlD* gene (without a frameshift) as well as the anaerobic B12 pathway. Again, the function of the *chlD* gene in these organisms was unclear, but the presence of the anaerobic B12 route may suggest that this pathway existed before the development of the CHL pathway.

To conclude, our analysis suggested a logical model for the coevolution of CHL and B12 pathways. We pinpointed specific prokaryotic species that can be considered as living fossils and their further study may shed a new light on the early evolution of these pathways. To our knowledge, the function of the *chlI* and *chlD* proteins without the large subunit (*chlH* or *cobN*) remained unexplored, and here, we suggested some model organisms where this could be studied.

## Figures and Tables

**Figure 1 microorganisms-10-01200-f001:**
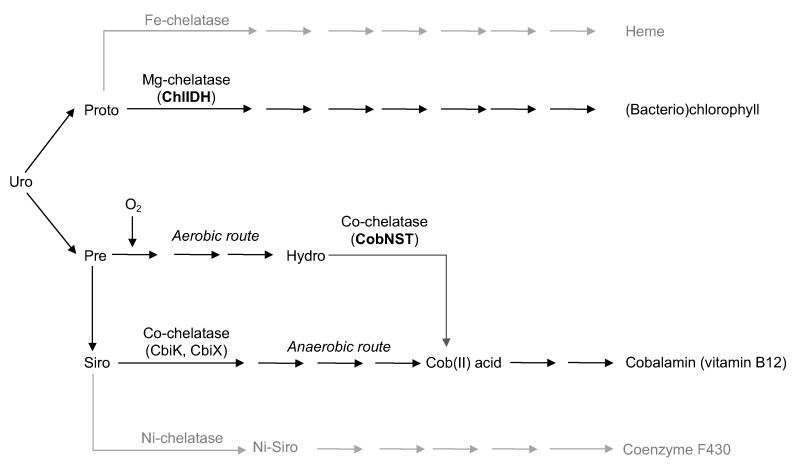
Overview of the existing biochemical pathways that produce (Bacterio)chlorophyll, Heme, Cobalamin, and Coenzyme F430 from Uroporphyrinogen III (Uro). The magnesium-chelatase ChlIDH inserts Mg^2+^ ion in the Protoporphyrin IX (Proto). There are two alternative pathways to synthesize Cobalamin. The CobNST chelatase inserts Co^2+^ in the aerobic path. It has been suggested that ChlIDH and CobNST have common evolutionary origin.

**Figure 2 microorganisms-10-01200-f002:**
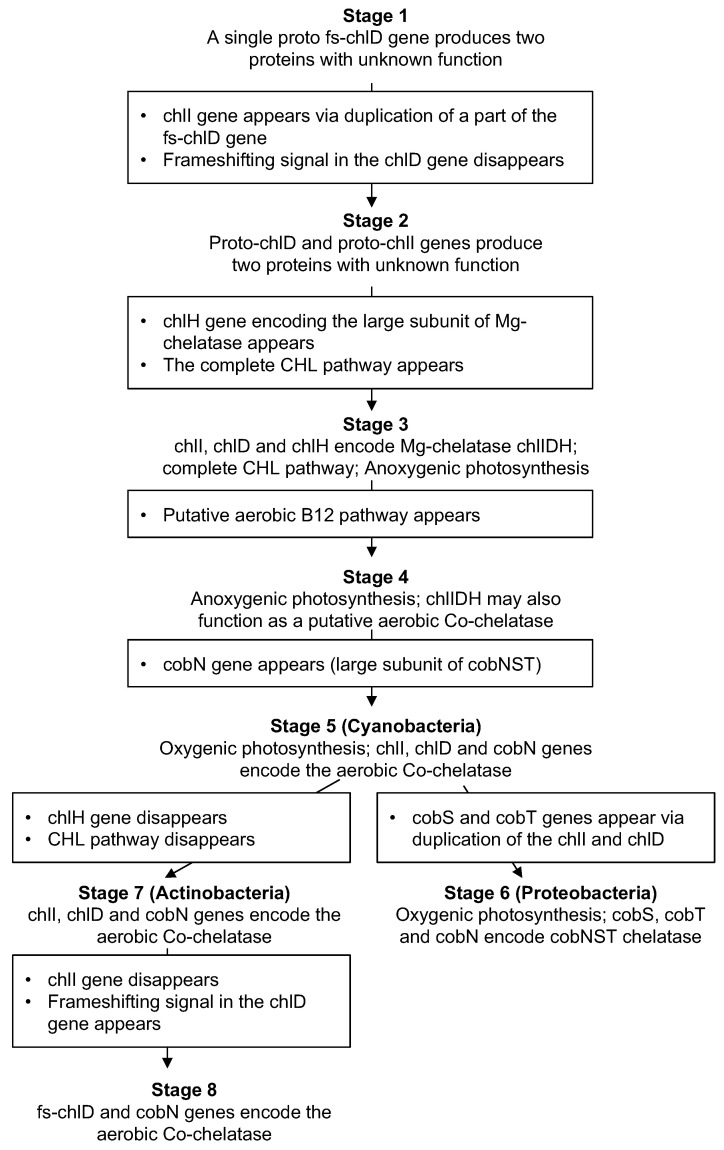
A putative chain of events that could happen between different stages during the evolution of the CHL and aerobic B12 pathways.

**Table 1 microorganisms-10-01200-t001:** The possible stages of the evolution of the genes encoding Mg-/Co-chelatases as well as CHL and B12 pathways.

Stage	Example Species (Phila)	Synthesis of the Medium and Small Proteins	Function of the Medium and Small Proteins	B12 Pathway(Anaerobic)	Large Subunit (chlH/cobN)	CHL Pathway	B12 Pathway(Aerobic)
**1**	*Spirochaeta thermophila* (Spirochaetes)*Brevefilum fermentans* (Chloroflexi)	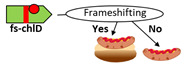	Unknown	-	-	-	-
**2**	*Thermodesulfobacterium commune* (Thermodesulfobacteria)*Thermodesulfovibrio yellowstonii* (Nitrospirae)	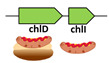	Unknown	-	-	-	-
**3**	*Polynucleobacter duraquae* (Proteobacteria)	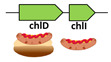	Mg-chelatase(small and medium subunits)	-	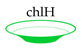	Yes	-
**4**	*Halorhodospira halophila* (Proteobacteria)*Thiocystis violascens* (Proteobacteria)	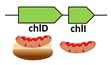	Mg-(Co-?)chelatase(small and medium subunits)	Yes	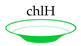	Yes	Yes?
**5**	*Synechococcus elongatus* (Cyanobacteria)*Prochlorococcus marinus* (Cyanobacteria)	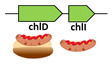	Mg- and Co- chelatases(small and medium subunits)	Yes	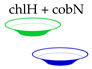	Yes	Yes
**6**	*Rhodospirillum rubrum* (Proteobacteria)*Dinoroseobacter shibae* (Proteobacteria)*Rhodovulum sulfidophilum* (Proteobacteria)	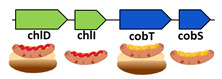	Mg- and Co- chelatases(small and medium subunits)	Yes	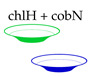	Yes	Yes
**7**	*Mycobacterium tuberculosis* (Actinobacteria)*Prauserella marina* (Actinobacteria)	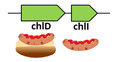	Small and medium subunits of Co- chelatases	Yes	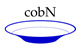	-	Yes
**8**	*Pseudomonas aeruginosa* (Proteobacteria)*Methanocaldococcus fervens* (Archaea)	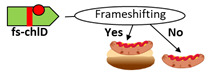	Small and medium subunits of Co- chelatases	Yes	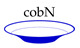	-	Yes

## Data Availability

The website of the database with all the information about 1207 prokaryotic genomes with the *chlD*, *bchD*, and *cobT* genes together with the corresponding information about the predicted frameshifts and frameshifting signals are available at http://ivanya.com/chelatase/ (accessed on 1 April 2022).

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
