# Peer review of "Translational Frameshifting in the chlD Gene Gives a Clue to the Coevolution of the Chlorophyll and Cobalamin Biosyntheses"

_microorganisms, 2022, doi:10.3390/microorganisms10061200_

Round 1

Reviewer 1 Report

The manuscript describes a possible evolutionary origin of the magnesium chelatase and aerobic cobalt chelatase small (ChlI/CobS) and medium subunits (ChlD/CobT). It also highlights the presence of these subunits in transcriptional units within genomes that are lacking large subunits (ChlH/CobN) of these chelatases and which lack the ability to synthesise either B12 or chlorophyll. The paper is short but to the point and conveys a clear message with data to support the conclusions.

Corrections of minor spelling errors such as "chlorohpyll" in the abstract and correct grammatical use of the the definite article, "the", need to be made. 

Author Response

We would like to thank the reviewer for positive feedback about our manuscript. We are glad that the main idea of our paper was clearly explained. The suggested corrections have been made.

Reviewer 2 Report

In this manuscript, Kuznetsov, Milenkin, and Antonov present their hypothesis on how the chlorophyll and aerobic cobalamin biosynthesis genes co-evolved through the bioinformatic analysis of genes encoding Mg-chelatases and aerobic Co-chelatases. This hypothesis is based on their previous identification of a frameshifting signal in a subset of chlD genes; a frameshift that allows for the production of two of the three subunits required for formation of an active Co-chelatase. Besides providing a predictive model on how the chlorophyll and aerobic  cobalamin biosynthesis genes co-evolved, the manuscript also introduces a chelatase database developed by the group. The manuscript was very well written and the predictive model is quite interesting, however, more data supporting the predictive stages of the model are needed to fully convince the reader. Some questions by this specific reviewer are:

  • Figure S2 indicates that 167 genomes have been found to possess fs-chlD genes, but only four taxa are mentioned in Table 1. While there is some rationale provided for the examples provided in Stage 1, what about stage 8?  Where do the fs-chlD genes from the other 163 genomes fall? Methanocaldococcus and Pseudomonas genera are not closely related, so is it possible that horizontal transfer played a role?
  • It is predicted that a duplication resulted in Stage 2. Do any genomes possess both two copies of fs-chlD or joint copies of fs-chlD and a chlI?
  • The chlH gene appears in Stage 3. Is there any data to suggest the origin of this gene?
  • In Figure S3 the bchD gene forms a monophyletic group separate from chlD. How does this affect the proposed model? The model focuses on the presence or absence of genes and metabolic ability, but what about simple protein phylogeny?
  • A reduction/simplification is proposed during stage 8 as the taxa provided do not produce chlorophyll, but do synthesize cobalamin. How does this fs-chlD differ (where is it located on the tree presented in Figure S3) from those identified in Spirochaeta and Brevefilum? Methanocaldococcus and Pseudomonas are in different domains, so it is difficult to imagine the same reduction/simplification occurred.
  • In regards to Supplemental Figure 3, is there any significance to the +1 versus -1 fs-chlD genes? Each type appears to group together on the tree? Can these genes be grouped by phyla?

Other minor edits:

The manuscript oscillates from italicizing gene names to no intalicizing them.

Line 37: change “swicth” to “switch”

Line 62: change “impleneted” to “implemented”

Line 108: change “phila” to “phyla”

Line 118: It might be better to refer to Supplemental Figure 5 instead of Figure 1.

Line 125: change “prokatyotic phila” to “prokaryotic phyla”

Lines 126-128: It would be helpful to describe a little more about the frameshift predicted to occur. Figure S4 is not very helpful.  While the “slippage signal” is marked, it is not clear how the secondary structure of the mRNA plays a role.

Line 155: change “anoxigenic” to “anoxygenic”

Line 190: It would be helpful to move the Supplemental Figure 5 into the main figures.

Author Response

We  thank the reviewer for carefully reviewing our work. We agree that some of our conclusions may need additional supporting data. Additional analysis have been performed to improve the quality of the manuscript (see below)

> Figure S2 indicates that 167 genomes have been found to possess fs-chlD genes,

> but only four taxa are mentioned in Table 1. While there is some rationale provided

> for the examples provided in Stage 1, what about stage 8?  Where do the fs-chlD

> genes from the other 163 genomes fall? Methanocaldococcus and Pseudomonas

> genera are not closely related, so is it possible that horizontal transfer played a role?

Table 1 in the present study only contains example species with appropriate sets of genes for demonstration purposes. Additional  details about all of the fs-chlD genes can be found in the ChelataseDB and have also been presented in our previous paper. (Antonov 2020).  https://academic.oup.com/mbe/article/37/8/2268/5811573

We agree with the reviewer, that horizontal gene transfer from Stage 1 to Stage 8 may be an appropriate assumption. Indeed, according to the phylogenetics tree, all the fs-chlD genes  do not cluster with wild-type chlD/bchD/Cobt genes. This indicates that the independent appearance of the frameshifting signal in the modern chlD genes seems to be unlikely. 

The following point has been added to the discussion section of the manuscript:

"The chlD genes with frameshifting signals in these genomes likely originated via horizontal gene transfer rather than the signals evolved once again in the conventional chlD/bchD genes. This is supported by the fact the fs-chlD genes do not group with the other genes without frameshifts (Supplementary Figure S3)"

> It is predicted that a duplication resulted in Stage 2. Do any genomes possess

> both two copies of fs-chlD or joint copies of fs-chlD and a chlI?

We would like to thank the reviewer for this logical question. Indeed the genotypes proposed by the reviewer can be expected between stage 1 and stage 2. We searched the ChelataseDB for such genomes and identified several potential candidates among Chloroflexi. We added the following text to the Results section:

"Moreover, we identified several Chloroflexi species (from the Anaerolineaceae group) with a potential intermediate genotype between Stage 1 and 2. Namely, Anaerolinea thermophila UNI-1, Ornatilinea apprima, Longilinea arvoryzae and Levilinea saccharolytica possessed the fs-chlD genes with characteristic frameshifting signals, but also had a separate chlI gene. Interestingly, these species were also strict anaerobes and thermophiles suggesting their ancient origin [29]. Thus, there were a number of living prokaryotes that provided insights into the very early evolution of the modern Mg-chelatase enzyme."

> The chlH gene appears in Stage 3. Is there any data to suggest the origin of this gene?

It is possible that the chlH gene encoding large chelatase subunit, also existed before the appearance of the Mg-chelatase and the chlorophyll pathway. The product of this gene may be able to perform a function that is not related to the chelatase. Indeed, we identified a number of genomes containing the chlH gene, but without genes encoding the medium or small chelatase subunits. Additional analysis needs to be performed to reveal this function.

> In Figure S3 the bchD gene forms a monophyletic group separate from chlD. How

> does this affect the proposed model? The model focuses on the presence or

> absence of genes and metabolic ability, but what about simple protein phylogeny?

bchD together with the bchI and bchH genes encode the Mg-chelatase bchIDH that are utilized by bacteria that produce bacteriochlorophyll. Clearly, the two Mg-chelatases (chlIDH and bchIDH) have common evolutionary origin, but the fact that the corresponding medium subunit genes form separate monophyletic groups may indicate their subsequent evolution was independent from each other. More details on this topic requires additional analysis that in our view is outside the scope of the current study.

> A reduction/simplification is proposed during stage 8 as the taxa provided

> do not produce chlorophyll, but do synthesize cobalamin. How does this fs-chlD

> differ (where is it located on the tree presented in Figure S3) from those

> identified in Spirochaeta and Brevefilum? Methanocaldococcus and

> Pseudomonas are in different domains, so it is difficult to imagine

> the same reduction/simplification occurred.

The idea about simplification from stage 5 (Cyanobacteria) to stage 7 (Actinobacteria without CHL biosynthesis) to stage 8 (fs-chlD gene) was our logical speculation based on the different genotypes that we observed. At the same time, the majority of the fs-chlD genes were present in the species that are able to synthesize B12 aerobically using the chimeric fs-chlD+cobN Co-chelatase. There were only a few examples of the fs-chlD genes that were present in the genomes without the cobN gene that we selected for Stage 1. Thus, the majority of the fs-chlD genes on the tree presented in Figure S3 corresponded to Stage 8. The details of the simplification in different prokaryotic taxa may need additional specialized analysis.

> In regards to Supplemental Figure 3, is there any significance to the +1

> versus -1 fs-chlD genes? Each type appears to group together on the tree?

> Can these genes be grouped by phyla?

The +1 frameshift was only observed in Actinobacteria, while -1 frameshifts were present in many different taxa (both Archaea and Bacteria). These two types of programmed frameshifting likely required different molecular mechanisms. At the same time, the bioinformatic identification of the -1 frameshifting signals was successful for many genomes and they are presented in the ChelataseDB. The detailed analysis of the -1 and +1 frameshifts in the chlD gene is available in our previous work (Antonov 2020).  https://academic.oup.com/mbe/article/37/8/2268/5811573

> Lines 126-128: It would be helpful to describe a little more about the frameshift

> predicted to occur. Figure S4 is not very helpful.  While the “slippage signal” is

> marked, it is not clear how the secondary structure of the mRNA plays a role.

The following text has been added to the Results section:

"In the chlD genes from various taxa -1 frameshifts have been observed more frequently than +1 frameshifts [16 ]. The recoding signals in these genes usually consisted of AT-rich "slippery sites" together with a stimulatory RNA secondary structures located downstream. More info about frameshifing mechanism can be found elsewhere [ 27 ]. Due to the stochastic nature of the programmed frameshifting, ribosomes reading the same mRNA can produce two different proteins that allows co-translational assembly of protein complexes"

Other minor edits:

> The manuscript oscillates from italicizing gene names to no intalicizing them.

All the gene names in the manuscript have been italicized.

> Line 37: change “swicth” to “switch”

> Line 62: change “impleneted” to “implemented”

> Line 108: change “phila” to “phyla”

> Line 118: It might be better to refer to Supplemental Figure 5 instead of Figure 1.

> Line 125: change “prokatyotic phila” to “prokaryotic phyla”

> Line 155: change “anoxigenic” to “anoxygenic”

> Line 190: It would be helpful to move the Supplemental Figure 5 into the main figures.

All the typos have been corrected

Round 2

Reviewer 2 Report

The authors have addressed my concerns.